# Targeting Protein-Protein Interactions to Inhibit Cyclin-Dependent Kinases

**DOI:** 10.3390/ph16040519

**Published:** 2023-03-31

**Authors:** Mark Klein

**Affiliations:** 1Hematology/Oncology Section, Primary Care Service Line, Minneapolis VA Healthcare System, Minneapolis, MN 55417, USA; mark.klein2@va.gov; 2Division of Hematology, Oncology and Transplantation, Department of Medicine, University of Minnesota, Minneapolis, MN 55455, USA

**Keywords:** protein-protein interaction inhibitor, cyclin-dependent kinase, cyclin, neoplasm

## Abstract

Cyclin-dependent kinases (CDKs) play diverse and critical roles in normal cells and may be exploited as targets in cancer therapeutic strategies. CDK4 inhibitors are currently approved for treatment in advanced breast cancer. This success has led to continued pursuit of targeting other CDKs. One challenge has been in the development of inhibitors that are highly selective for individual CDKs as the ATP-binding site is highly conserved across this family of proteins. Protein-protein interactions (PPI) tend to have less conservation amongst different proteins, even within protein families, making targeting PPI an attractive approach to improving drug selectivity. However, PPI can be challenging to target due to structural and physicochemical features of these interactions. A review of the literature specific to studies focused on targeting PPI involving CDKs 2, 4, 5, and 9 was conducted and is presented here. Promising lead molecules to target select CDKs have been discovered. None of the lead molecules discovered have led to FDA approval; however, the studies covered in this review lay the foundation for further discovery and develop of PPI inhibitors for CDKs.

## 1. Introduction

Cyclin-dependent kinases (CDKs) are serine/threonine kinases that are involved in several cellular processes, including regulation of the cell cycle, neuronal function, and transcription [1]. In general, CDKs are inactive until bound to a partner cyclin and undergo conformational changes to phosphorylate or interact with target proteins [1]. Specifically, CDKs 4 and 6 play a particularly important role in priming phosphorylation of retinoblastoma protein (RB), and as a result, result in propagation of the cell cycle through the G1/S checkpoint [1,2,3]. CDK4/6 inhibitors that are FDA approved for treatment of patients with advanced breast cancer include palbociclib, abemaciclib, and ribociclib [4,5,6]. Trilaciclib is a CDK4/6 inhibitor approved for use in bone marrow support for patients with small cell lung cancer undergoing treatment with myelosuppressive chemotherapy [7].

CDKs are likely best recognized regarding the cell cycle propagation [1,2]. Cyclin D1 binds CDK4 or CDK6, which subsequently phosphorylates RB on a single site to prime RB for further phosphorylation [8]. CDK2 is then activated by Cyclin D1 and further phosphorylates RB, allowing multiple transcription factors to become active and drive cell cycle progression [1,2]. The CDK inhibitor and tumor suppressor p16^INK4a^ inhibits CDKs 4 and 6; however, its loss, via deletion or promoter methylation, may allow unchecked cell cycle progression and promote cancer progression in several tumor types, such as breast cancer, esophageal cancer, head and neck cancer, mesothelioma, non-small cell lung cancer, and pancreatic cancer [1,2,3,9]. The CDK inhibitor p16^INK4a^ (p16, a protein) inhibits the Cyclin D1-CDK4 protein complex [1,2,3]. The inactivation of p16 promotes the pathogenesis of many tumor types, including mesothelioma, breast cancer, pancreatic cancer, non-small cell lung cancer, esophageal cancer, and head and neck cancer [1,2,3,9]. Since CDK4/6 inhibitors have made a significant impact on clinical outcomes in advanced breast cancer, targeting of other CDKs may lead to improved clinical outcomes in various cancers. However, the development of very selective inhibitors of individual CDKs has not yet been achieved. Part of this may be due to the similar structural biology of the ATP-binding sites that are the targets for competitive inhibitors of CDKs [10]. Targeting protein-protein interactions (PPI) is a strategy that, while challenging, holds promise in discovery and development of more selective inhibition of CDK activity [11]. As opposed to targeting the ATP-binding site, PPI targeting occurs outside the enzymatic site, predominantly between CDKs and accompanying cyclins.

## 2. Biology and Structure of CDKs

Upwards of 20 different CDKs have been discovered [12]. Here, we focus on CDKs that may have applicability to cancer treatments (Table 1). In normal cells, CDK2 plays a significant role in cell cycle propagation by phosphorylating retinoblastoma protein (RB) as a key event in propagation through the G2/M checkpoint [1]. CDK4 is introduced previously. CDK5 (associated with activating partners p25, p29, p35, and p39) has been associated with the following cellular activities: transcriptional and translational regulation, microtubule activity, cytoskeletal activity, cell adhesion, the cell cycle, and DNA damage [13]. CDK9, when complexed with Cyclin T in the P-TEFb complex, plays a major role in RNA transcription, namely in elongation [14].

CDKs generally consist of (1) an N-terminal lobe containing an anti-parallel β-sheet and a (2) C-terminal lobe of mostly α-helices connected by a hinge region, where the ATP-binding site is in the cleft between the two lobes [15]. An activation loop in the N-lobe consists of an α-helix that interacts with the C-helix and affects the back-side of the active site cleft, and a glycine rich loop plus the C-helix in the N-lobe forms a PSTAIRE sequence that are both important to CDK activity [12,15]. These regions are highly conserved amongst CDKs [11]. However, cyclins have much less sequence homology, so the interaction between CDKs and cyclins may be exploited to introduce more selective inhibitors [11].

## 3. Examples of PPI Inhibition in CDKs

### 3.1. CDK2/Cyclin A

CDK2 requires binding to the Cyclin A or Cyclin E family of cyclin proteins [11]. To date, efforts to target PPI for CDK2 has focused on the interactions between CDK2 and Cyclin A [16,17,18]. Much of the effort in this area has focused on a strategy to target either the p21Waf naturally occurring inhibitor of CDK2 or the cyclin binding groove located on CDK2 [11,19]. The strategy called REPLACE (**R**eplacement with **P**artial **L**igand **A**lternatives using **C**omputation **E**nrichment) has resulted in lead molecules that disrupt CDK2-Cyclin A interactions and a decrease in CDK2 enzymatic activity [16,17,18,19]. The strategy involves (1) identifying and/or determining the three-dimensional (3D) structure of a lead peptide in contact with the cyclin binding groove, (2) identify structural and chemical features of the peptide-protein interactions, (3) truncate the peptide starting at the N-terminus, (4) use computational techniques to identify small-molecule alternatives to the truncated peptide (partial ligand alternatives, PLAs) that retain key binding characteristics to the target CDK, (5) identify commercial sources for or synthesize small molecule alternatives (PLAs), (6) synthesize fragment-ligated peptides (FLIP) by ligating the small molecule alternative (PLA) to the truncated peptide, (7) test the efficacy of the candidate FLIPs in an in vitro binding assay and/or functional assay, and (8) evaluate the efficacy of the candidate FLIPs in a cell viability assay [19]. Figure 1 shows an example of this process.

An example of this approach to identify inhibitors of the CDK2/Cyclin A interaction is the discovery of benazmide capped peptidomimetics [16]. Previously, an octapeptide inhibitor, HAKRRLIF, derived from the protein p21Waf1 that naturally interacts with CDK2, binds the cyclin binding groove on CDK2 [20,21]. Key PPI include KRR-CDK2 ion pairs and hydrophobic side chain interactions [20,21]. Previous studies had demonstrated that replacing the N-terminal tetrapeptide with groups to (1) interact with a hydrophobic pocket or an arginine-binding site (but not both) resulted in successful replacement groups (Figure 1) [17,18,22,23]. Iterations of synthesis and structure/function studies where monosubstituted and 3,4-substituted benzoic acid derivatives were substituted as caps to yield peptide mimetics yield promising lead molecules [16]. The most effective monosubstituted inhibitor included a guanidinomethyl group at the 4-position in a binding assay (IC_50_ = 0.69 μM for CDK2/Cyclin A, Table 2, top) [16]. Addition of a hydroxyl group to the 3-position of the benzoic acid derivative resulted in an IC_50_ = 4.56 μM while a 3,4-substitution with a methoxy group at the 3-position and a piperidinyloxy group at the 4-position result in a slightly worse binding affinity (IC_50_ = 22.42 μM) (Table 2, middle and bottom). These groups are consistent with the hypothesis that enhancing leads to bind the hydrophobic pocket or arginine-binding site result in increased affinity to the cyclin binding groove. However, when evaluated in cell viability assays (U2OS sarcoma and DU145 prostate cancer cells), the compounds evaluated had little to no efficacy (IC_50_ > 150 μM) with regard to decreasing cell viability. It is unclear what properties preclude these compounds from being active in these cancer cells.

In a separate study, cavity analysis was conducted on the complex of CDK2/Cyclin A to identify druggable pockets [24]. Two druggable pockets outside of the ATP-binding site and next to the activation site at the interface of CDK2 and Cyclin A were identified (druggable pockets 1 and 2 in Figure 2). An in silico screen with 1925 FDA-approved drugs was conducted and the top 10 hits were evaluated in vitro and in vivo. The drug homoharringtonine demonstrated high affinity for CDK2, inhibited acute myeloid leukemia cell growth in vitro, induced autophagic degradation, and demonstrated activity against a leukemia mouse model.

### 3.2. CDK4/Cyclin D

To date, no small-molecule (i.e., <500 molecular weight) PPI inhibitors have been discovered that target CDK4/Cyclin interactions. However, computational analysis of the CDK4/Cyclin D interface has been performed to see how mutations affect PPI [25]. Non-synonymous single nucleotide polymorphisms (nsSNPs) can result in changes to the physicochemical properties of individual amino acid residues [25]. As SNPs are quite common, complementary computational methods (SIFT, PolyPhen 2, and I-Mutant 3.0) to predict phenotypic effects of SNPs were applied to the three-dimensional structure of CDK4/Cyclin D1 (PDB ID: 2W96) [25]. SIFT results in a tolerance score for residue replacement, PolyPhen 2 uses evolutionary and physical considerations to predict how a change in amino acid affects protein structure and function, and I-Mutant 3.0 results in prediction of protein stability based on differences in Gibbs free energy differences between the natural and variant proteins.

Of the nsSNPs evaluated, SIFT predicted 11, PolyPhen 2 predicted 9, and I-Mutant predicted that 15 nsSNPS, respectively, to be deleterious to CDK4/Cyclin D1. When taking all the data into account, 5 nsSNPs were predicted to be highly deleterious across platforms corresponding to the following mutations: R24C, Y180H, A205T, R210P, and R246C. These were then used for further computational analysis. When comparing the number of predicted hydrogen bonds predicted during molecular dynamics simulations between CDK4 and Cyclin D1 in the native structure and after applying the above mutations from nsSNPs, the maximum number of hydrogen bonds was predicted to decrease for each of the mutant proteins, suggesting a potential decrease in stability of the complex. In addition, the predicted minimum distance between CDK4 and Cyclin D1 was predicted to be higher in the mutant complexes than in the native complexes. Finally, the solvent-accessible surface area was minimally different between complexes (ranging from ~18.5 nm^2^ to ~22 nm^2^ for the mutant complexes to ~18.5 nm^2^ to ~21.5 nm^2^ for the native complex). Overall, the predicted changes due to nsSNPs could be utilized as part of a PPI inhibitor strategy to take advantage of pre-existing changes in complex stability.

While not truly a small molecule (i.e., <500 molecular weight), efforts to develop structurally constrained peptides to target CDK4 have been described [26]. A significant aspect of the interface involves the interaction between the PISTVRE sequence from the C-helix from CDK4 and the a5 helix from Cyclin D1 (Figure 3). A series of peptides derived from the PISTVRE sequence and adjacent residues were designed and synthesized (Table 3 and Figure 3 and Figure 4). Residues L49, I51, V54, R55, V57, and R61 from CDK4 were identified as critical for interactions along the interface, so residues corresponding to S52, E56, L60, and E64 were identified as being sites for substitutions for introducing olefin-based rings. The peptide P2short exhibited the highest affinity for CDK4. P2short, and a derivative, P2shortA, inhibited A549 lung cancer cells in vitro with IC_50_ values at approximately 10 μM. Intravenous injection in mice with A549 xenografts with P2shortA did not result in targeting lung tumors.

### 3.3. CDK5/p25

The interface between CDK5/p25 was evaluated to identify PPI inhibitors as potential lead molecules [27]. Figure 5 shows the interface between CDK5/p25, where the C helix from CDK5 is hypothesized to be critical to the CDK5/p25 interaction [27,28]. Computational alanine scanning was conducted by replacing residues from the C helix one at a time in separate computational experiments: L49, R50, I52, C53, and L54. Calculated binding free energies were predicted to have the most effect on by L49A, C53A, and R50A, respectively in order. These residues formed the core structure template for a pharmacophore that was used for virtual screening of the National Cancer Institute (NCI) diversity database. The molecules identified via virtual screening predicted to binding most tightly to p25 centered around these three residues. A CDK5/p25 kinase assay and a BRET-based assay in yeast were utilized to evaluate ability of the lead compounds to inhibit or bind to CDK5/p25 [29]. Compound NSC88915 from the NCI database exhibited the most activity against CDK5/p25 with an IC_50_ = 6 μM. None of the compounds exhibited binding activity in the yeast model, suggesting that the leads likely did not penetrate the yeast cells.

### 3.4. CDK9/Cyclin T

CDK9 is activated upon binding by Cyclin T and the resulting activated complex plays a core function of transcriptional regulation as part of the positive transcription elongation factor b (P-TEFb) complex [30]. While some CDK9 inhibitors, such as dinaciclib, have been developed, another strategy to disrupt this activity is to prevent CDK9/Cyclin T interactions [30,31,32]. Strategies to develop PPI inhibitors to disrupt CDK9 and Cyclin T interactions have focused on molecular docking, molecular dynamics, small-molecule synthesis and accompanying structure–activity relationship (SAR) studies, and peptide design [31,32,33,34].

In one study, the CDK9/Cyclin T interface was analyzed via molecular dynamics and docking techniques and a series of peptides were designed to disrupt the interaction [33]. Peptides were chosen as a starting point to design lead molecules as no hydrophobic pockets were evident in the analysis. Hot spot analysis was used to guide where to focus peptide design where residues L93, G137, and F146 from Cyclin T and residue F12 from CDK9 contribute the majority of the predicted binding energy of the CDK9/Cyclin T interaction. Four peptide fragments from Cyclin T (PDB ID:3BLH) were chosen corresponding to amino acids 89–94, 137–142, 141–146, and 141–148. In addition, an unbiased de novo peptide was used to identify additional peptides (six additional peptides) that may be lead molecules. The peptides were computationally docked with CDK9 and analyzed via molecular dynamics to rank the peptides with the highest predicted affinity for CDK9 by predicted free energy of binding. Computational procedures were conducted with CDK9 in complex with the P-TEFb complex and as a free protein. The peptides LQTLFGEL (corresponding to amino acids 89–94) and ESIILQ (corresponding to amino acids 141–146) from Cyclin T exhibited the highest predicted binding (lowest free energy) to CDK9.

In a separate study, the same researchers conducted further analyses of the CDK9/CyclinT interface by utilizing the FTMap server to conduct fragment-based (organic probe) prediction of structural clusters where probes may bind [34,35]. Three of the four predicted clusters identified on the CDK9 interface surface overlapped with the LQTLF peptide fragment (part of LQTLFGEL above). No such sites were identified on the Cyclin T surface. Cyclin T residues L6 and F146 were utilized to define a pharmacophore based on FTMap-based mapping and from the previous peptide studies [33,35]. The subsequent initial pharmacophore included five points from L6 and F146 features: an aromatic core from F146, two hydrogen bond donors from L6, one hydrogen bond acceptor from the carbonyl of F146, and a hydrophobic point from the carbons of the L6 side chain. Virtual screening with AnchorQuery (chosen for its optimization for PPI inhibitors) yielded three hits [36]. The pharmacophore was then modified by excluding the hydrophobic point from the L6 side chain, and 26 hits were obtained via AnchorQuery, and 2 compounds were identified via ZincPharmer [37]. The hits docked with CDK9 and analyzed after application of molecular dynamics simulations. A derivative of 2-amino-8-hydroxyquinoline was the highest ranked lead.

Two studies have identified leads to disrupt CDK9/Cyclin T interactions and demonstrate activity against breast cancer cells [31,32]. CDK9 may be a good target in breast cancer due to overexpression of CDK9 in this cancer [31,32]. In the first study, the CDK9/Cyclin T 3-dimensional structure (PDB ID:6GZH) was used for virtual screening of the ZINC database, and the top 14 hits were evaluated for inhibition of CDK9/Cyclin T activity via a chemiluminescence assay. IC_50_ values of 3–6.6 μM were determined for the most effective compounds via this assay. Enzyme kinetics analysis revealed that the lead compound (a derivative of tetrahydroisoquinoline) was a competitive inhibitor to ATP (K_i_ = 2.14 +/− 0.2 μM). However, a co-immunoprecipitation experiment with the lead compound in MDA-MB-231 breast cancer cells demonstrated that the CDK9/Cyclin T interaction was disrupted. This lead compound demonstrated cytotoxicity against the following breast cancer cell lines: MDA-MB-231 (IC_50_ = 6.2 μM), MDA-MB-468 (IC_50_ = 7.4 μM), and BT549 (IC_50_ = 12.6 μM).

In the second study, it was hypothesized that based on some of the above research that a quinolone scaffold as part of an organometallic complex may have potential to disrupt PPI, so a library of metal complexes that included 7-chloro-2-phenylquinoline CN as the template structure was constructed [32]. In order to study SAR, the number and position of methyl groups were varied. The library was subsequently screened in a chemiluminescence assay of CDK9/Cyclin T activity. At a screening concentration of 10 nM, the most potent compound exhibited 55.9% inhibition compared to 31.0% inhibition exhibited by the control compound dinaciclib. The EC_50_ for the most potent compound was 3.3 nM. Other compounds evaluated were less potent than the top hit. SAR demonstrated that methyl location that differed from the top compound resulted in decreased CDK9 inhibitory activity. Enzymatic studies confirmed the lead compound was non-competitive with ATP and exhibited K_i_ = 3.3 nM. Co-immunoprecipitation experiments in MDA-MB-231 breast cancer cells demonstrated that the compound decreased CDK9/Cyclin T interactions. Activity against CDKs 1, 2, 4, 6, 5, and 12 were not significantly inhibited by the most potent compound, suggesting that his lead has high selectivity for CDK9/Cyclin T. The IC_50_ for the top compound demonstrated cytotoxicity against the following cell lines: MDA-MB-231 (210 nM), 4T1 (400 nM), MCG-10A (702 nM), and MCF-7 (690 nM). As CDK9 regulates oncogenes c-Myc and Mcl-1, immunoblotting experiments were conducted after compound exposure in MDA-MB-231 cells, and both proteins (c-Myc and Mcl-1) exhibited decreased expression. Finally, treatment of a mouse allograft model with a murine 4T1 breast cancer cell line with the most potent compound resulted in a significant decrease in tumor volume after a 13-day treatment.

## 4. Discussion

CDK4/6 inhibitors have proven to be highly effective in advanced breast cancer [4,5,6]. To date these inhibitors, palbociclib, abemaciclib, and ribociclib, have been classical small molecule ATP-competitive inhibitors. However, CDK4/6 inhibitors have yet to result in major clinical efficacy in other cancers and targeting other CDKs have yet to result in clinical applications. The studies summarized in this review point to early successes in developing lead molecules for identification of new principles in targeting PPI and introducing selectivity to CDK inhibitory activity. To date, most of the work towards development of targeting CDK/cyclin PPI has been focused on CDK2 and CDK9 activity. There has not been a systematic approach to targeting PPI across the family of CDKs and partner cyclins. The clinical success of CDK4 inhibitors and the early success in targeting PPI provide the framework for further evaluating the promise of targeting PPI in the CDK family of proteins.

## Figures and Tables

**Figure 1 pharmaceuticals-16-00519-f001:**
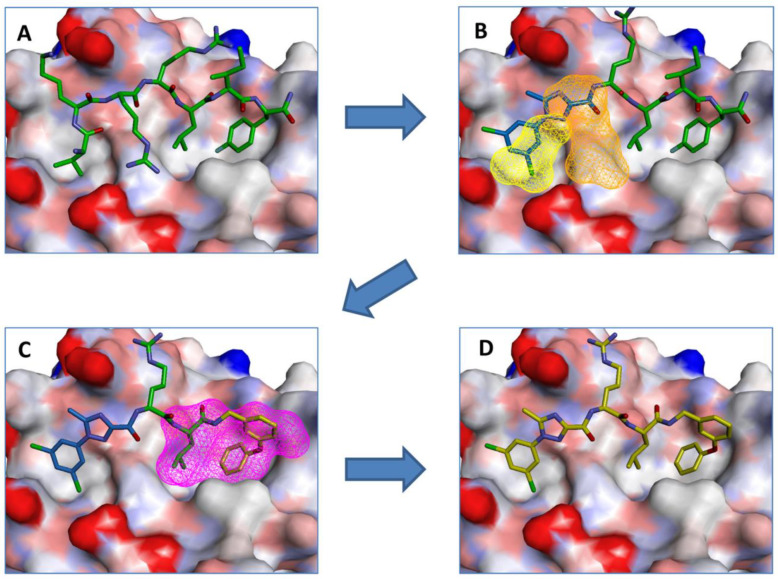
Example of the REPLACE protocol utilized to convert the peptide sequence HAKRRLIF to an N-capped and C-capped dipeptide (PDB ID: 2UUE). (**A**–**C**). Peptide HAKRRLIF docked to a hydrophobic pocket. (**D)**. The modified peptide. Used with permission [17].

**Figure 2 pharmaceuticals-16-00519-f002:**
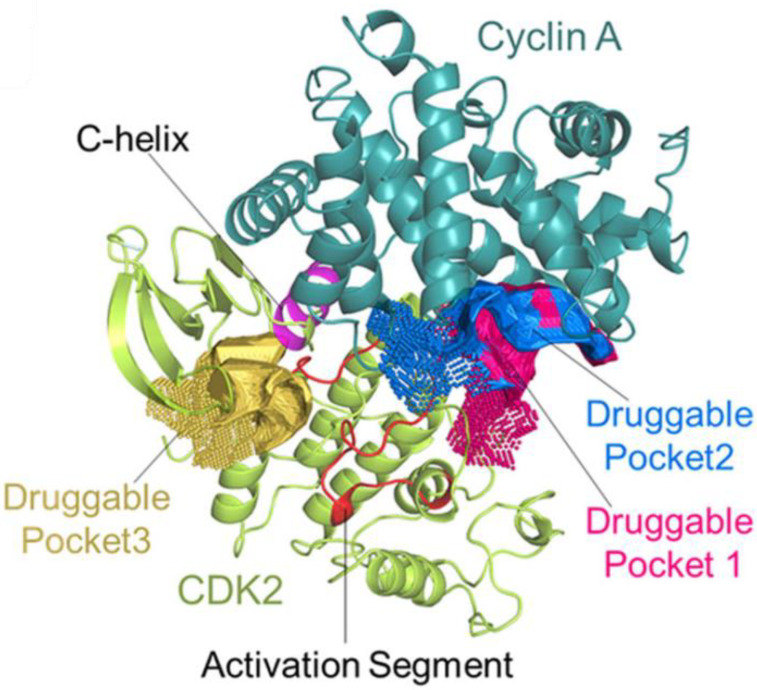
The 3D structure of CDK2/Cyclin A with druggable pockets labeled with permission [24].

**Figure 3 pharmaceuticals-16-00519-f003:**
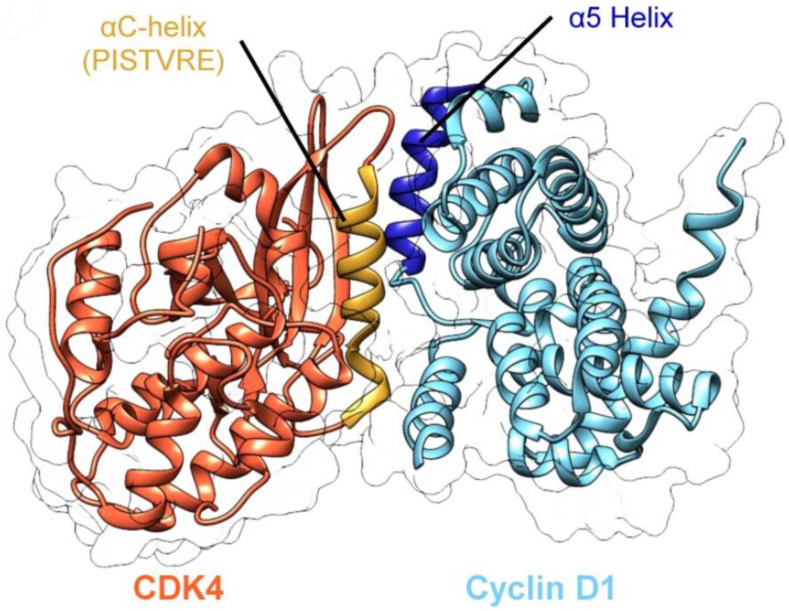
The overall 3D structure of CDK4/Cyclin D1 is shown with the protein–protein interface helices labeled. Adapted with permission [26].

**Figure 4 pharmaceuticals-16-00519-f004:**
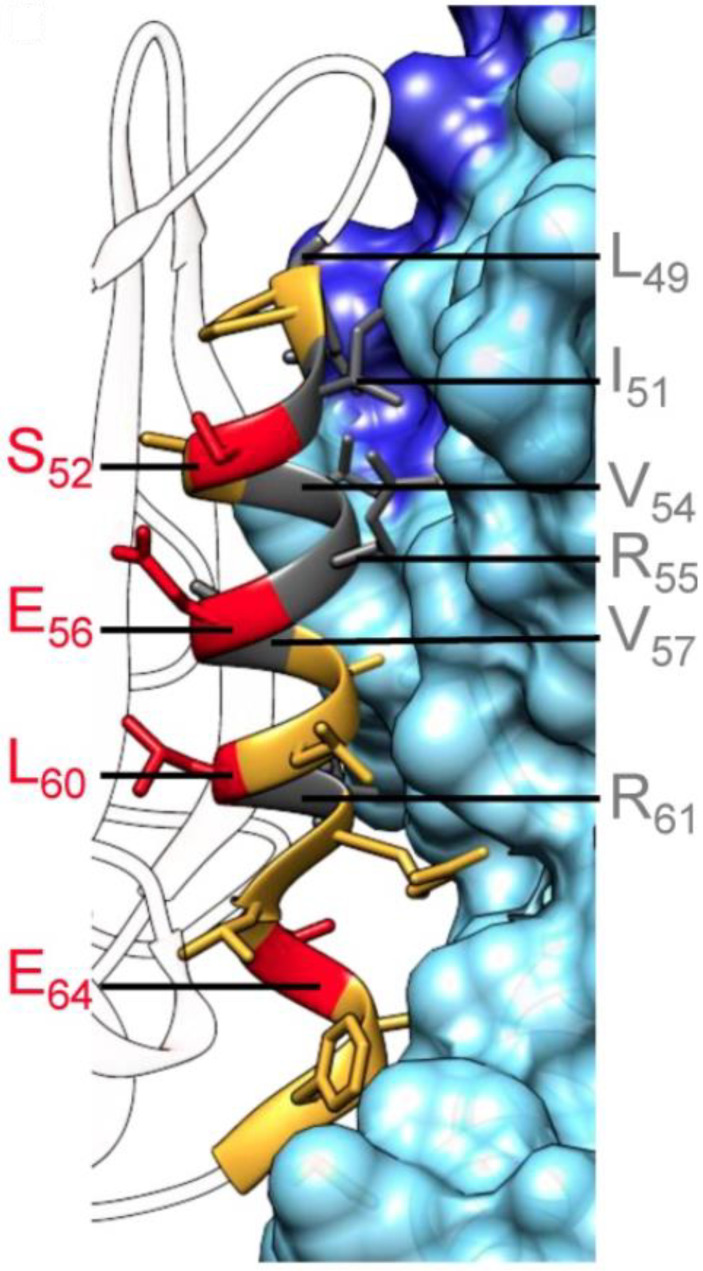
CDK4/Cyclin D1 is shown with the protein–protein interface helices labeled. Adapted with permission [26].

**Figure 5 pharmaceuticals-16-00519-f005:**
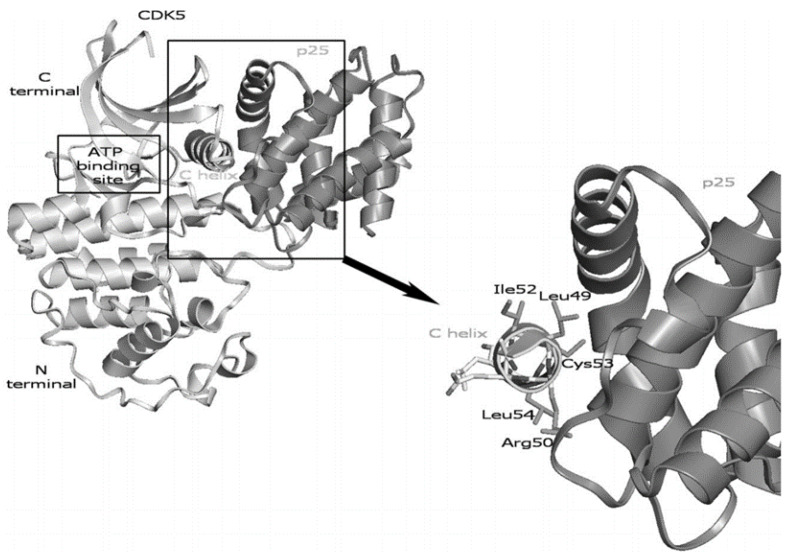
The overall 3D structure of CDK5/p25 is on the left. The right side of the figure depicts a closeup of the C helix bound to p25. Adapted with permission [27].

**Table 1 pharmaceuticals-16-00519-t001:** Biologic roles of key CDK/Cyclin pairs.

Protein Complex	Key Biologic Role	FDA-Approved Drugs
CDK2/Cyclin A	Cell Cycle	None
CDK4/Cyclin D and CDK6/Cyclin D	Cell Cycle	palbociclib, abemaciclib, ribociclib, trilaciclib
CDK5/p25, CDK5/p35 CDK9/Cyclin T	Neuronal function Transcription	None None

**Table 2 pharmaceuticals-16-00519-t002:** Examples of key CDK2/Cyclin A inhibitor lead compounds. From [16] with permission.

Scaffold	R1	R2	R3	R4
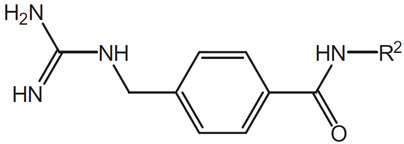	H	RLNpfF		
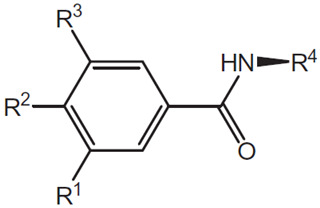	OH	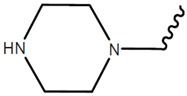	H	RLIF
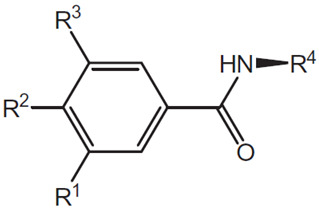	OCH_3_	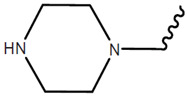	H	RLIF

**Table 3 pharmaceuticals-16-00519-t003:** Select peptides from CDK4 as noted in the text from [26]. S5 denotes site for olefin staple anchor.

Name	Peptide Sequence
aC-helix (native from CDK4)	PISTVREVALLRRLEAFE
P2	Ac-GLPISTVRS_5_VALS_5_RRLEAFE-NH_2_
P2short	Ac-CLPISTVRS_5_VALS_5_RRL-NH_2_
P2shortA	ALPISTVRS_5_VALS_5_RRL-NH_2_

## Data Availability

Not applicable.

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
