# Peer review of "Targeting Protein-Protein Interactions to Inhibit Cyclin-Dependent Kinases"

_pharmaceuticals, 2023, doi:10.3390/ph16040519_

Round 1

Reviewer 1 Report

This paper is a review of PPI inhibitors targeting CDKs. Although there was a previous review by the same author, the inhibitors for each target protein are summarized compactly, so it is considered that there is no problem with publication in this journal.

Author Response

Thank you for your review.

Reviewer 2 Report

This review covers an important topic that has not been extensively reviewed elsewhere—inhibitors of CDKs that work by inhibiting their protein-protein interactions with cyclins.  The review as it is very thin and not that interesting to read, but ultimately will be worthy of publication provided the following items are fixed.

1.       The figures are very randomly chosen and many more need to be added.  This review would be much improved if each section started with a structure of the CDK/cyclin complex, highlighting the interfacial region, similar to what is done with CDK5/p25 (although make it in color!).  Then the structures of the molecules (peptides or otherwise) should be given as separate figures.

2.       A number of computational tools are described in the discovery process, but most are defined with a single sentence (an exception is REPLACE in Fig. 1).  More discussion about how the tools work in the context of these inhibitors is important.

3.       There aren’t that many PPI inhibitors to these targets, so every effort should be made to include all of the studies.  A fairly quick search led to a notable omission (Zhang, J. et al.  Nat. Commun. 13, 2835 (2022), and there are likely others that should be discussed.

Author Response

This review covers an important topic that has not been extensively reviewed elsewhere—inhibitors of CDKs that work by inhibiting their protein-protein interactions with cyclins.  The review as it is very thin and not that interesting to read, but ultimately will be worthy of publication provided the following items are fixed.

  1. The figures are very randomly chosen and many more need to be added. This review would be much improved if each section started with a structure of the CDK/cyclin complex, highlighting the interfacial region, similar to what is done with CDK5/p25 (although make it in color!).  Then the structures of the molecules (peptides or otherwise) should be given as separate figures.

Response: Thank you for the helpful comments. The review is considered an “Opinion” for Pharmaceuticals, so it is more narrow in scope compared to a more extensive review. Efforts have been made to add more structures; however, there is some limitation to what figures are available, per copyright restrictions. I initially felt that placing structures into the manuscript that are not from the authors may confuse readers and distort the message from the individual papers. The CDK5/p25 figure was in black and white in the original manuscript as published. I have added a figure for CDK2 and CDK4. In addition, a few key small molecule structures were added.

  1. A number of computational tools are described in the discovery process, but most are defined with a single sentence (an exception is REPLACE in Fig. 1). More discussion about how the tools work in the context of these inhibitors is important.

Response: A paragraph on REPLACE is included. However, it seems that the details of each computer program would be a bit out of the scope of an Opinion.

  1. There aren’t that many PPI inhibitors to these targets, so every effort should be made to include all of the studies. A fairly quick search led to a notable omission (Zhang, J. et al.  Commun. 13, 2835 (2022), and there are likely others that should be discussed.

Response: I initially did a literature search (including PubMed) for representative papers. In trying to hold more true to small molecule inhibitors, some of these recommendations were left out. I added information as referenced above and the article from Theranostics 2020 p2008.

Reviewer 3 Report

The review covers the interesting topic of targeting PPIs to inhibit cdks. The section on the biology of the cdks is lacking in detail, a figure at this point is usual to explain the role of the cell-cycle and non-cell-cycle cdks and their partner cyclins. This detail is also important to establish the likely differences between an ATP-competitive cdk inhibitor and a cdk-cyclin PPI inhibitor.

The section with examples of PPIs is rather uneven. A good amount of useful detail is given to the cdk2/cyclinA inhibitors. Although 2D chemical structures of key compounds would be helpful here. Also, discussion of the lack of cellular activity of the compounds should be include with respect to common medicinal chemistry rules.

Key references for cdk2/A inhibitors are missing e.g. Nature 2022 p7571. No discussion is given to cdk2/E inhibitors despite the literature in this area.

The section on cdk4/D inhibitors is missing key reference: Theranostics 2020 p2008. The details given require an illustration of the structures used and a map of the mutations.

The discussion section is very brief and lacks a critical analysis of the state of the field in context with the clinical status of the ATP-competitive inhibitors.

Overall, the work is rather patchy and inconsistent, key references have been missed, and it lacks detailed analysis. As such it is not publishable in its current form and requires an extensive rewrite with significant additional material.

Author Response

  1. The review covers the interesting topic of targeting PPIs to inhibit cdks. The section on the biology of the cdks is lacking in detail, a figure at this point is usual to explain the role of the cell-cycle and non-cell-cycle cdks and their partner cyclins. This detail is also important to establish the likely differences between an ATP-competitive cdk inhibitor and a cdk-cyclin PPI inhibitor.

Response: A brief introduction to the difference between competitive inhibitor and PPI inhibitor was placed in the introduction.

  1. The section with examples of PPIs is rather uneven. A good amount of useful detail is given to the cdk2/cyclinA inhibitors. Although 2D chemical structures of key compounds would be helpful here. Also, discussion of the lack of cellular activity of the compounds should be include with respect to common medicinal chemistry rules.

Review: A few key figures of structures were added. However, the compounds are currently lead compounds, so in the framework of an opinion I felt that adding extensive discussion of Rules of 5 by Lipinski is a bit out scope in the early stage of this field.

  1. Key references for cdk2/A inhibitors are missing e.g. Nature 2022 p7571. No discussion is given to cdk2/E inhibitors despite the literature in this area.

Response: This CDK2/A inhibitor was included as an example in the field.

  1. The section on cdk4/D inhibitors is missing key reference: Theranostics 2020 p2008. The details given require an illustration of the structures used and a map of the mutations.

Response: The article was included in the revision.

  1. The discussion section is very brief and lacks a critical analysis of the state of the field in context with the clinical status of the ATP-competitive inhibitors.

Overall, the work is rather patchy and inconsistent, key references have been missed, and it lacks detailed analysis. As such it is not publishable in its current form and requires an extensive rewrite with significant additional material.

Response: The article is an opinion and not a review, so it is of a more narrow scope than a full review. We added a fair amount of information as above to increase the scale within the scope of an Opinion.

Round 2

Reviewer 3 Report

The author has made some additional and modifications to the manuscript in line with the comments. The review presents the most of the current activity in the field in a readable format.

There are a number of minor errors that require correcting before the manuscript is ready for publication.

p2 ln 50 change back to CDK4/6
p4 table 2 the structures need to be scaled to a sensible size, column widths need adjusting so that CH3 isn't split across 2 lines
p4 looks like the reference numbering is out of sequence
p6 ln 143 correct to 'performed'
p7 ln 171 correct to 'series'
p8 and p9  poor resolution of images
p10 onwards references are in italics
p11 ln 222 correct to 'residues', ln 224 correct to 'energy', ln 257 change 'identified' to 'determined'
p12 ln 289 change 'classic' to 'classical'

Author Response

I have made changes to the following suggestions as requested:

p2 ln 50 change back to CDK4/6 - done
p4 table 2 the structures need to be scaled to a sensible size, column widths need adjusting so that CH3 isn't split across 2 lines - done
p4 looks like the reference numbering is out of sequence – references doublechecked and ok
p6 ln 143 correct to 'performed' - done
p7 ln 171 correct to 'series' - donep10 onwards references are in italics - done
p11 ln 222 correct to 'residues', ln 224 correct to 'energy', ln 257 change 'identified' to 'determined' - done
p12 ln 289 change 'classic' to 'classical' - done

Regarding the p8 and p9 poor images, I adjusted the image 8 in the manuscript. The figure on p9 was obtained with permission, so that is not editable.
